# Advances in Femtosecond Laser GHz-Burst Drilling of Glasses: Influence of Burst Shape and Duration

**DOI:** 10.3390/mi14061158

**Published:** 2023-05-30

**Authors:** Pierre Balage, Guillaume Bonamis, Manon Lafargue, Théo Guilberteau, Martin Delaigue, Clemens Hönninger, Jie Qiao, John Lopez, Inka Manek-Hönninger

**Affiliations:** 1Université de Bordeaux-CNRS-CEA, CELIA UMR 5107, 33405 Talence, France; 2AMPLITUDE, Cité de la Photonique, 33600 Pessac, France; 3Chester F. Carlson Center for Imaging Science, Rochester Institute of Technology, Rochester, NY 14623, USA

**Keywords:** ultrafast laser processing, femtosecond GHz-bursts, laser–material interaction, glasses, percussion drilling

## Abstract

The femtosecond GHz-burst mode laser processing has attracted much attention in the last few years. Very recently, the first percussion drilling results obtained in glasses using this new regime were reported. In this study, we present our latest results on top-down drilling in glasses, focusing specifically on the influence of burst duration and shape on the hole drilling rate and the quality of the drilled holes, wherein holes of very high quality with a smooth and glossy inner surface can be obtained. We show that a decreasing energy repartition of the pulses within the burst can increase the drilling rate, but the holes saturate at lower depths and present lower quality than holes drilled with an increasing or flat energy distribution. Moreover, we give an insight into the phenomena that may occur during drilling as a function of the burst shape.

## 1. Introduction

Femtosecond laser material processing in the so-called GHz-burst mode has attracted much attention in the last few years. The first investigations of this laser–matter interaction regime published by the group of Ömer Ilday [1,2] were mainly focused on metals’ and semiconductors’ ablation, and showed a remarkable increase in the removal rate compared to the single-pulse regime; this was confirmed shortly after proving the accumulative thermal regime of the process [3], and has been further investigated in a systematic study of ablation rates, including a comparison of the GHz-burst mode with single-pulse femtosecond as well as nanosecond laser ablation from the literature [4]. All these papers state that the GHz burst regime allows for an ablation rate comparable to nanosecond laser processing, but with the quality of femtosecond laser processing, which could represent significant progress for the laser micromachining industry. Indeed, the pulse-to-pulse delay within the GHz-burst involves timescales in the order of magnitude of the nanosecond, which is the same as the heat relaxation time. Other studies show a lower ablation efficiency for GHz-bursts than for multiple single pulses, depending on the burst parameters, such as laser fluence and number of pulses per bursts [5,6,7]. In these contributions, the authors attest to an important screening effect induced by the timescales involved in the GHz-burst regime (around 30 ns). However, in these experiments, bursts containing a very restricted number of pulses with rather high energy were applied. Therefore, in these studies the use of GHz-bursts showed less efficiency, and was even partly detrimental to the ablation quality. Thus, in such cases, the utility of the burst mode is very limited, as the energy repartition of the pulses within the burst is not optimized, and the beneficial accumulative effect cannot be exploited.

Until now, only a few studies have focused on milling experiments in dielectric materials [8,9,10]. These contributions report all an improvement of the ablation rate, but at the expense of the processing quality. Indeed, these three publications point out the fact that the ablation rate is highly dependent on the number of pulses per bursts and increases with an increasing number of pulses within the burst. A comparative study of fused silica ablation with repetitive single pulses, MHz bursts and GHz bursts confirms these findings [11]. Very recently, we demonstrated top-down percussion drilling of glass materials and the associated process dynamics using in situ monitoring [12,13] and thermal imaging [13]. Furthermore, we have shown that it is possible to achieve efficient drilling even for pulses with sub-threshold fluences, thanks to thermal accumulation within the GHz-burst.

Several methods were investigated for glass drilling, including the use of CO_2_ lasers [14] and Bessel beams [15]. Some techniques have produced high-quality and high-aspect-ratio holes by means of UV technologies [16,17,18]. Furthermore, very recently, huge potential has been foreseen in GHz burst micromachining in an interesting review article [19], and even more recently, an experimental study supported this potential by attesting to the possibility of drilling high-aspect-ratio crack-free holes in glasses using this regime [20].

In this study, we report the influence of burst duration and shape on the drilling rate and hole geometry. We tested four different burst shapes, defined as the classical burst, the decreasing burst, the increasing burst and the flat burst, which correspond to different energy repartitions of the pulses within the bursts. We also studied the evolution of the hole depth as a function of the number of bursts applied on the sample, as well as the quality of the inner walls of the drilled holes.

## 2. Materials and Methods

Our experiments were carried out using an industrial laser system, a modified Tangor 100 from Amplitude, which is described in reference [13] and produces pulses of 530 fs pulse duration at a wavelength of 1030 nm. This flexible laser system allows for a precise optimization of the drilling parameters such as burst repetition rate, burst energy, number of pulses per burst, and burst shape. In this study, we investigated the influence of the number of pulses within the burst as well as the burst shape on the drilling performance. From a practical point of view, the gain depletion in the laser amplifier leads to an uneven intensity profile during the burst, as shown in Figure 1, both schematically (a), and as a screen shot from the measured photodiode signal using an oscilloscope (photodiode EOT-3500 and oscilloscope MSO70404C, from Tektronix) (b). This phenomenon can be pre-compensated by applying a tuned pulse energy distribution on the burst before amplification to obtain the desired burst shape after the amplification [21]. This technique allows us to design three different burst shapes, as depicted in Figure 1, in addition to the classical burst shape: the decreasing burst shape (c,d), the increasing burst shape (e,f) and, finally, the flat burst shape (g,h). The number of pulses per burst was set to 36, 70, 100, 130, and 160, respectively, for the classical burst shape, and to 50 and 100 for the three others.

We chose these burst shapes for the following reasons. The decreasing burst shape was chosen in order to determine if higher energy pulses in the beginning of the burst would optimize the heating process of the material and thus the drilling process. The increasing burst shape was also implemented to determine if lower pulses in the beginning of the burst could increase the overall quality of the hole by more slowly heating the material and keeping a reasonable ablation rate, with the high energy pulses at the end of the burst. The flat burst was investigated as a compromise between the two first configurations. The intra-burst repetition rate was set to 1.28 GHz for all experiments, creating bursts of several nanoseconds. These parameters allow for reaching a time scale of the same order of magnitude as the heat diffusion time, and allow for controlled thermal accumulation throughout the burst duration [22]. For these experiments, we chose a rather low burst repetition rate, 1 kHz, since a previous study [13] has shown detrimental side-effects on some glasses for repetition rates above 10 kHz, notably in sodalime, for which high repetition rates can lead to the appearance of a large heat-affected zone surrounding the hole. At even higher repetition rates, a new phenomenon can arise, leading to a counter-effect, as the glass might reach the softening temperature and collapse under its own weight, thereby closing the just drilled hole. Moreover, the glass might also suffer from chemical decomposition and gas release accompanied by the appearance of bubbles due to high temperature.

The holes were drilled by focusing a Gaussian beam on the surface of the different glasses, thanks to a microscope objective Mitutoyo Plan NIR Apo 5X, resulting in a measured spot size of 9.3 µm (1/e^2^ diameter) and an effective numerical aperture of 0.14. The spot size was measured using a homemade calibrated magnification system with an uncertainty of ±0.64 µm. Thanks to a top view Basler CMOS camera, a white light and a couple of dichroic mirrors, we can visualize through the focusing objective and accurately set the position of the laser focus at the front surface of the different glass samples, as can be seen in Figure 2. During the drilling, we used a side-view system composed of a green diode emitting at 520 nm for illumination, and a Basler camera (Basler acA1920-25mu) coupled with a long-distance microscope (InfiniMax KX with MX-6 Objective) for real-time imaging. The latter is equipped with a 520 nm bandpass filter in order to visualize directly through the samples, and not to be blinded by the processing laser wavelength. The focusing head is mounted on a Z-motorized stage (VP25X, MKS Instruments), whereas the sample is fixed on a motorized XY-monolithic stage (One-XY60, MKS Instruments).

The XYZ-stages and the laser gate are controlled by DMCpro software (Direct Machining Control, Vilnius, Lithuania). The latter is equipped with an automatic find focus function, allowing for a precision in the focus position down to a few microns, thereby reducing the positioning uncertainty and leading to the same conditions for all drilling experiments. The workstation has a granite base and gantry, ensuring a high stability and an excellent repeatability of the experiments. An optical measuring microscope (MF-B1010D, Mitutoyo) is used for ex situ imaging, achieving high accuracy measurements of the hole depths and diameters with a precision of ±2.2 µm + 0.02·L, with L being the measured length in mm.

## 3. Results and Discussion

### 3.1. Influence of the Burst Duration

We studied the evolution of the hole depth as a function of the number of bursts applied to the samples of sodalime and fused silica in a range from 1 burst to 10,000 bursts. The resulting holes for the two materials, using a classical burst shape with 300 bursts, 400 bursts, and 500 bursts, are depicted in Figure 3, for a burst energy of 172 µJ, different numbers (30, 70, 100, and 130) of pulses per bursts (ppb) at a burst repetition rate of 1 kHz, and an intraburst repetition rate of 1.28 GHz.

These images were taken with a 20X microscope. During this study, we observed that the position of the focus is a critical point. If the focalization is located even less than 10 µm underneath the surface, the resulting hole presents cracks, and the surface of the sample is highly affected. So, care has to be taken to accurately position the laser focus on the sample surface. The holes on these images present all an excellent quality without exception. Moreover, one can clearly observe a very different morphology of the holes obtained with standard repetitive single-pulse femtosecond technology [13]. Indeed, the holes here are cylindrical for both materials and the inner surface is very smooth, even very glossy in fused silica. In addition, one may note the absence of a heat-affected zone surrounding the holes. In sodalime, we distinguish some structures of the inner walls, but the tip remains smooth. These images attest that the number of pulses within the burst does not affect the general morphology of the hole. However, in a range from 300 to 500 bursts, we can observe that for a higher number of pulses per burst, the depth seems to increase, which confirms the results found in Ref. [10]. This observation attests that for bursts with more pulses, the accumulative regime is enhanced, and the energy usage is optimized.

The graphs corresponding to the evolution of the hole depth throughout the full range of drilling time are presented in Figure 4. The uncertainty in the depth measurements is not displayed on this graph, as the values were negligible (lower than ±5 µm). This figure shows that the behavior of the two materials regarding the number of bursts is quite similar, leading to comparable hole depths, as published in [20]. Moreover, from these graphs, we identify for all configurations the same three distinct stages of the drilling process: (1) surface ablation, (2) confined ablation, and (3) ablation termination, where the drilling depth saturates, as explained in Ref. [20]. Figure 4 clearly shows that for both the classical and flat burst shapes, the number of pulses per burst as well as the energy repartition within the burst have a direct impact on the drilling rate and on the maximum achievable hole depth. Indeed, the zooms on Figure 4 show an increase in the drilling rate for higher numbers of pulses per burst. However, the hole depth saturates at lower values for increasing numbers of pulses per burst.

We note that for bursts of more than 70 pulses, the energy of each pulse within the burst is lower than the ablation threshold for both materials, and yet, hole drilling is still observable. The latter point attests that this particular ablation regime relies on thermal accumulation, thanks to the high intraburst repetition rate. Furthermore, we observe that the ablation depth at saturation is higher for the flat burst with 50 ppb than the one obtained with the classical burst shape containing 36 ppb, which attests to a better use of the energy. The drilling rates and the maximum hole depth extracted from the depth measurements (slope) for a burst energy of 172 µJ are summarized in Table 1, with Rate 1 corresponding to surface ablation and Rate 2 corresponding to confined ablation before saturation. The confined ablation rate is significantly lower than the surface ablation rate, probably implying a screening effect due to a denser ablation plume in the case of confined ablation. Indeed, the plume can expand freely in the air in the first tenths of microns of the drilling process, resulting in a low-density plume, but it will get denser as soon as the hole goes deeper, wherein the plume starts interacting with the inner walls of the hole. The surface ablation rate doubles in sodalime when the number of pulses per burst increases from 30 to 160, which confirms the observations made in Refs. [3,10]. Regarding fused silica, the same tendency in drilling rate increase is observed, and an interesting point can be seen at 160 pulses per bursts.

For this particular configuration, there was no clear drilling visible, as the individual pulse energy within the burst might not be high enough due to the higher ablation threshold of fused silica, and despite the accumulative regime. For a burst energy of about 200 µJ, a clear drilling was observed for a classical burst containing 160 pulses (results not presented in this paper). Interestingly, the maximum drilling depths reached in both materials are of the same order of magnitude, despite the fact that fused silica presents a much higher bandgap than sodalime (3.9 eV for sodalime [23], 9.0 eV for fused silica [24]), and thus a higher ablation threshold (2.9 J/cm^2^ for sodalime, 3.6 J/cm^2^ for fused silica [25]). We suppose that this phenomenon appears as a consequence of the difference of the inner wall quality of the holes. Indeed, fused silica presents glossy walls with a high reflection coefficient and very few scattering losses, which allows for a better transmission of the laser beam towards the tip of the hole.

### 3.2. Influence of the Burst Shape

In this part, we present the difference between the increasing burst, the decreasing burst and the flat burst on the drilling process in sodalime. Note that the classical burst shape was not investigated in this part, as its behavior is pretty close to the decreasing burst, leading to the same results as can be seen in Figure 1. We fixed the number of pulses per burst at 100 and the burst energy at 200 µJ. We note that the burst shaping process requires a certain number of pulses to be relevant; for shorter bursts, the burst shaping had no visible effect, as it requires a certain number of pulses on the rising edge and on the falling edge. The resulting holes are shown in Figure 5. These images were taken with a 10X objective mounted on the optical microscope. In Figure 5, we can once again observe holes with an excellent quality for the increasing (b) and the flat (c) burst shapes. It appears that for the decreasing burst configuration, the depth increases faster than for the two other configurations. However, for this burst shape, the holes present some cracks surrounding the inner walls, which is in full agreement with the observations of the cavity millings reported in [8,9,10,11], with bursts of high energy pulses.

A graphical representation of the depth as a function of the number of bursts applied to the sample for the three configurations is depicted in Figure 6.

This graph demonstrates that the burst shape obviously has a clear impact on the drilling dynamics. The drilling rate for decreasing bursts is much higher than for the two other configurations. However, we also observe that the saturation depth is lower. The drilling rates extracted from this graph as well as the average saturation depths for each configuration are summarized in Table 2.

This table shows that the surface ablation rate (Rate 1) is much higher for the decreasing configuration than for the other two configurations. This can be explained by the fact that the first pulses of the burst carry more energy than those of the two other configurations. Therefore, the heating of the materials from the first pulses will be too quick, which will result in a drilling process quite similar to the MHz-burst regime and thus a high ablation rate. On the other hand, higher intensity at the beginning of the burst may lead to a higher screening effect induced by a faster accumulation of plasma, resulting in a saturation that appears sooner in that case. Moreover, the relatively high energy of the first pulses within the burst does not allow for fully profiting from the accumulative GHz-burst regime, which is similar to ablation experiments involving bursts containing a small number of high-energy pulses [5,6,7]. Indeed, it appears that an increasing burst shape allows for deeper hole drilling. This can be explained by the fact that the low-energy pulses slowly heat the material and generate ablation incubation and low screening effects, while the high-energy pulses arriving at the end of the burst will improve the material ejection from the hole. Finally, we can see that a flat burst shape allows for even deeper hole drilling, but at the expense of the ablation rate (Rate 2). This configuration appears to be a good compromise, since it benefits from the advantages of both the previous burst shapes. The relatively low energy of the first pulses slowly heats the material, generating a low screening effect, and the relatively high energy of the last pulses induces an efficient ablation and thus deeper holes.

Figure 7 shows the images of the holes with full depth for a drilling time of 1 s (i.e., 1000 bursts applied). They were obtained by stitching several images measured by the 50X microscope, which has a limited field of view. These images confirm that the hole quality depends on the burst shape configuration, despite applying comparable burst energies. In the case of a decreasing burst shape, the hole is much deeper than for the two other configurations; this is the case because with 1000 bursts applied, none of the three burst shapes reaches the saturation depth. However, we can clearly see multiple cracks appearing, and the overall morphology is heading towards MHz burst drilling [26]. This indicates that a higher energy from the first few pulses may initiate the creation of the cracks.

Regarding the two other burst shapes, the hole morphology and depth are very similar. Note that the increasing burst shape seems to produce fewer cracks, and thus the best hole quality, probably because the very first pulses carry and deposit a low amount of energy.

## 4. Conclusions

The influence of the number of pulses per burst and the burst shape with an increasing number of bursts were investigated for top-down percussion drilling in glasses. This study revealed that the number of pulses per burst does not significantly affect the morphology and quality of the holes. However, short bursts result in deeper holes, whereas a higher number of pulses per burst produces a higher drilling rate, but at the expense of the depth of the hole. Moreover, the burst energy must be increased for long bursts in order to overcome the drilling threshold in the accumulative regime, which depends on the material. We furthermore investigated the influence of the burst shape on the drilling results with three different burst shapes, the decreasing, the flat, and the increasing burst, while keeping the same burst energy. In other words, the energy repartition on the pulses within the burst was varied. This study showed that a decreasing burst shape presents a higher surface ablation rate than the other burst shapes, allowing for fast drilling at the expense of the hole quality. This burst shape is preferable for limited depth drilling and tolerating a certain quality degradation. On the contrary, a flat burst shape presents the lowest surface ablation rate but a good hole quality, and allows for reaching the greatest hole depths. This burst shape should be used for applications requiring maximum drilling depth and good quality. Finally, the increasing burst shape should be implemented as a compromise solution, because it presents a better quality for deep holes and a noticeably higher ablation rate than the flat burst, but at the expense of the ability to reach maximum hole depths. In this study, we have presented different laser parameters, such as the number of pulses per bursts and the energy repartition within the burst, and showed that they lead to different drilling results; their precise controlling allows us to obtain holes of known depth and quality. Overall, this study constitutes a significant contribution to our collective understanding of laser processing using the GHz-burst regime, and proves its interesting potential for industrial precision micromachining of dielectric materials.

## Figures and Tables

**Figure 1 micromachines-14-01158-f001:**
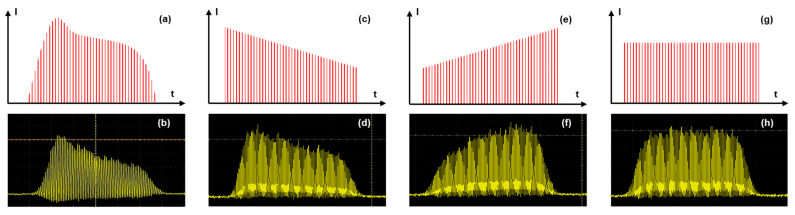
Schematic representation and measured shape of a classical burst (**a**,**b**), of a decreasing burst (**c**,**d**), of an increasing burst (**e**,**f**), and of a flat burst (**g**,**h**).

**Figure 2 micromachines-14-01158-f002:**
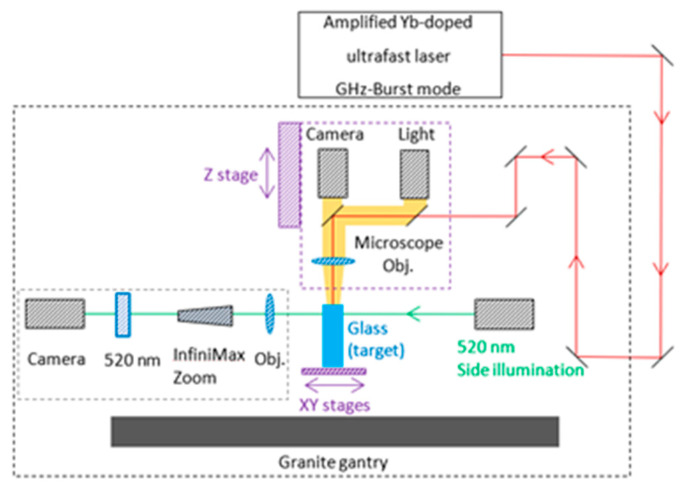
Blueprint of the experimental setup used for the drilling experiments.

**Figure 3 micromachines-14-01158-f003:**
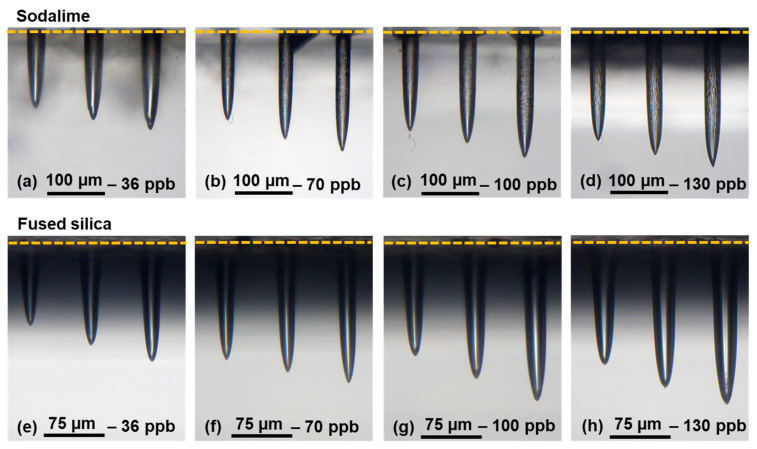
Microscope images of the holes drilled with a classical burst shape and a burst energy of 172 µJ in sodalime for 300 bursts, 400 bursts, and 500 bursts with 36 ppb (**a**), 70 ppb (**b**), 100 ppb (**c**) and 130 ppb (**d**), and for fused silica for bursts of 36 ppb (**e**), 70 ppb (**f**), 100 ppb (**g**) and 130 ppb (**h**).

**Figure 4 micromachines-14-01158-f004:**
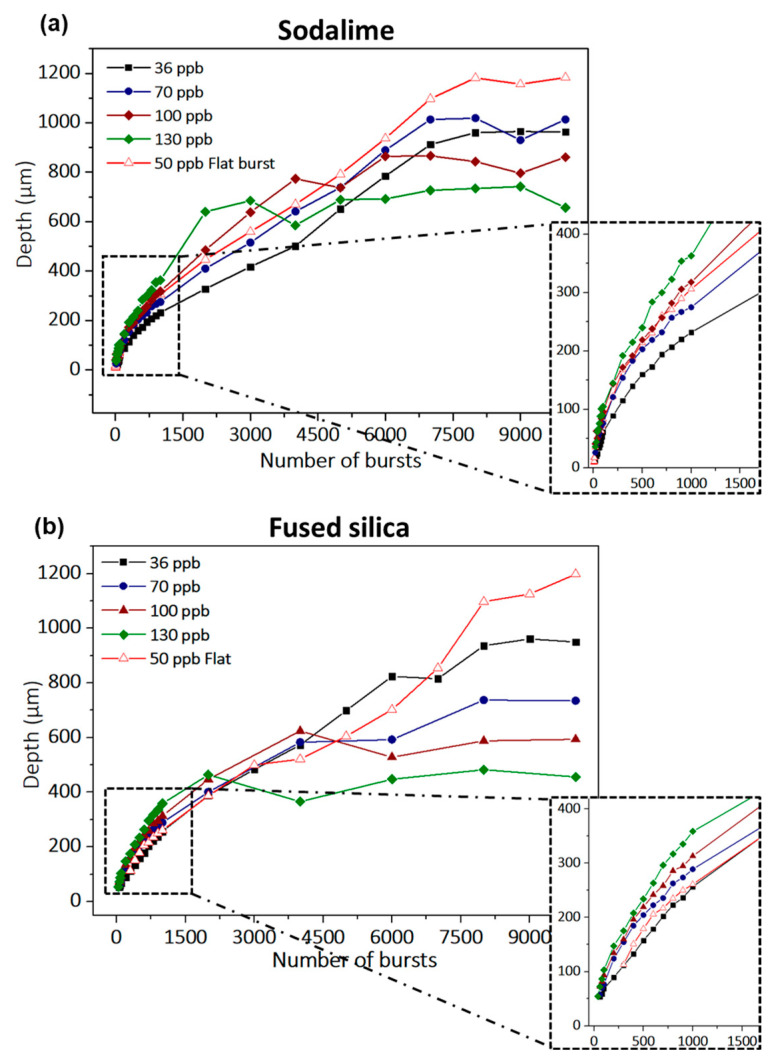
Evolution of the hole depth as a function of the number of bursts up to 10,000 bursts of 172 µJ burst energy for sodalime (**a**) and fused silica (**b**). The inserts on the right bottom corner are a zoom of the very first part of the graphs delimited by a rectangle of black dashed lines.

**Figure 5 micromachines-14-01158-f005:**
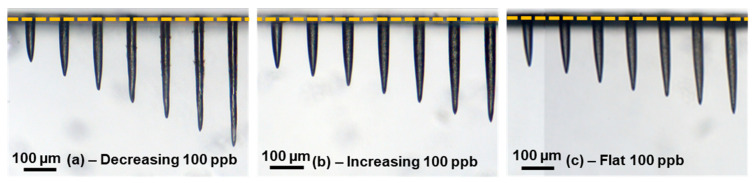
Microscope images of the holes drilled in sodalime with 100-pulse GHz-bursts of 200 µJ, with a number of bursts in a range from 200 to 800 for a decreasing burst shape (**a**), for an increasing burst shape (**b**) and for a flat burst shape (**c**).

**Figure 6 micromachines-14-01158-f006:**
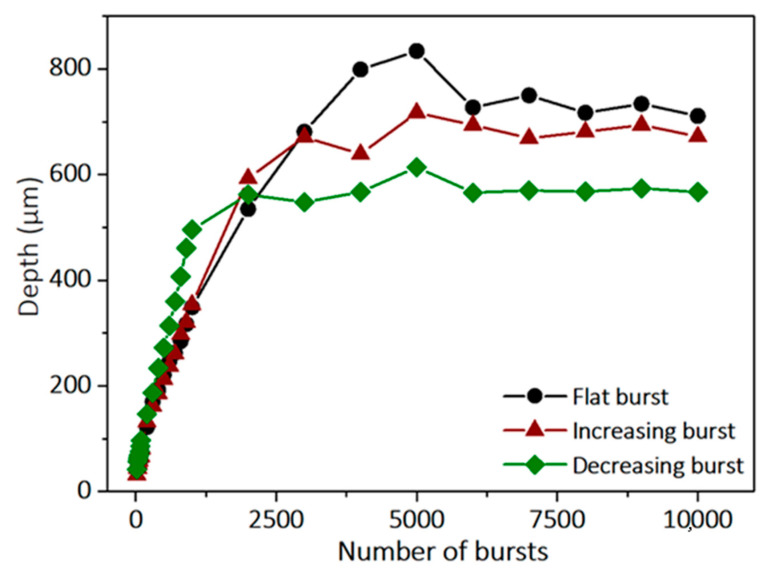
Evolution of the hole depth in sodalime as a function of the number of bursts applied on the sample for the three burst shapes, with bursts of 100 pulses and a burst energy of 200 µJ.

**Figure 7 micromachines-14-01158-f007:**
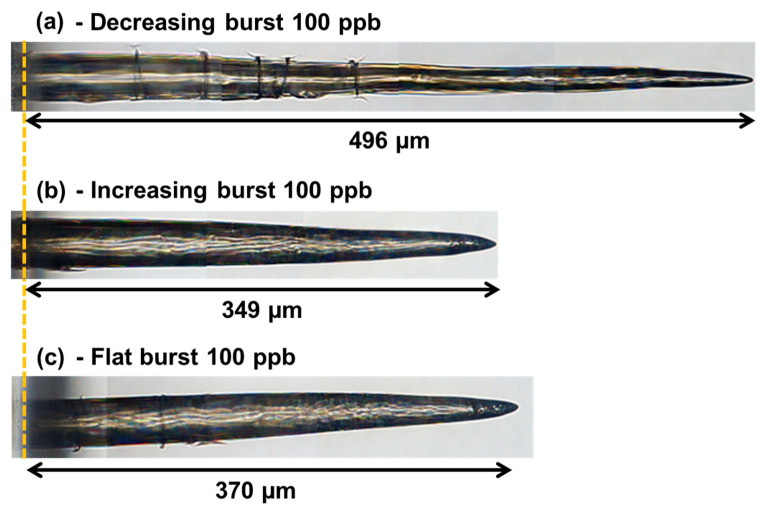
Microscope image of the holes drilled in sodalime for a drilling time of 1 s, with bursts of 100 pulses at a burst energy of 200 µJ, and a decreasing burst shape (**a**), an increasing burst shape (**b**), and a flat burst shape (**c**).

**Table 1 micromachines-14-01158-t001:** Drilling rates and maximum achievable depth in sodalime and fused silica for a burst energy of 172 µJ for the classical burst shape (36 ppb to 160 ppb), and the flat burst shape with 50 pulses per burst.

	Sodalime			Fused Silica		
Burst	Rate 1 (µm/Burst)	Rate 2 (µm/Burst)	Maximum Depth (µm)	Rate 1 (µm/Burst)	Rate 2 (µm/Burst)	Maximum Depth (µm)
36 ppb	0.65	0.11	950	0.23	0.11	1000
70 ppb	0.7	0.12	1000	0.48	0.10	800
100 ppb	0.9	0.16	820	0.6	0.12	600
130 ppb	1.2	0.28	690	0.78	0.27	420
160 ppb	1.3	0.35	550	/	/	/
50 ppb Flat	0.5	0.12	1190	0.3	0.12	1180

**Table 2 micromachines-14-01158-t002:** Drilling rates and maximum depth reached for decreasing, flat and increasing 100-pulse GHz-bursts, with a burst energy of 200 µJ.

Laser Parameters	Rate 1 (µm/Burst)	Rate 2 (µm/Burst)	Maximal Depth (µm)
100 ppb decreasing	1.2	0.45	614
100 ppb increasing	0.6	0.25	717
100 ppb flat	0.6	0.16	834

## Data Availability

Data underlying the results presented in this paper are not publicly available at this time, but may be obtained from the authors upon reasonable request.

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
