# Peer review of "Advances in Femtosecond Laser GHz-Burst Drilling of Glasses: Influence of Burst Shape and Duration"

_micromachines, 2023, doi:10.3390/mi14061158_

Round 1
Reviewer 1 Report
The authors suggest manuscript titled “Advances in femtosecond laser GHz-burst drilling of glasses: influence of the burst shape and duration”. The work is rigorously performed, and in my opinion is suitable for Micromachines. The authors carefully described their research and the manuscript is well structured. The comparison of the effect of different burst shapes and pulses per burst looks interesting and useful for the community.
I think, for completeness, the authors could mention some relevant theoretical studies, e.g., in the introduction.
Author Response
Dear reviewer,
We would like to thank you very much for your careful reading of our manuscript, your very positive feedback and your suggestion to include more references in the introduction. We improved the introduction considerably and added more references on experimental studies in the field as our work represents experimental work.
With best regards and on behalf of all the authors,
Inka Manek-Hönninger and Pierre Balage
Reviewer 2 Report
The submitted paper is within the scope of the journal, but there are several points which needed to be justified for the reader's point of view.
a) Abstract should contain results of the author's work. It should be summarized well.
b) Introduction section, the proposed method, materials and application scenarios used need to be described more clearly. Each cited document should be presented separately.
c) Experiments, The micro-structures after drilling, if it can be photographed. This can be used as a discussion of the physical phenomenons of different materials and different drilling conditions.
Quality of Presentation is GOOD.
Author Response
Dear reviewer,
We would like to thank you very much for your careful reading of our manuscript and your valuable suggestions for its improvement.
Point a) Abstract should contain results of the author's work. It should be summarized well.
Our response: Thank you very much for the remark. We added a short summary of the results in the abstract (see red-lined version).
Point b) Introduction section, the proposed method, materials and application scenarios used need to be described more clearly. Each cited document should be presented separately.
Our response: Thank you very much for the suggestions. We considerably improved the Introduction section by adding some references and describing them in more details and more separately where it was possible (to avoid repeating). Moreover, we added some information in the Materials and Methods mentioning important points for application scenarios (see red-lined revised version).
Point c) Experiments, The micro-structures after drilling, if it can be photographed. This can be used as a discussion of the physical phenomenons of different materials and different drilling conditions.
Our response: Thank you very much for the comment. We added a "Discussion" to the Results section and give important information on the differences in sodalime and fused silica drilling (see red-lined revised version).
Figure 7 shows the drilled holes taken with a 50x microscope objective which is the best resolution we can provide for showing the structures of the drilled holes.
We checked the English editing, and some minor typos were corrected (see red-lined revised version).
Best regards on behalf of all the authors,
Inka Manek-Hönninger and Pierre Balage
Round 2
Reviewer 2 Report
Author's has been revise and corrected reviewer's comments. I suggestion this manuscript can be accepted for publication to the micromachines journal in this status.